# AR Splicing Variants and Resistance to AR Targeting Agents

**DOI:** 10.3390/cancers13112563

**Published:** 2021-05-23

**Authors:** Mayuko Kanayama, Changxue Lu, Jun Luo, Emmanuel S. Antonarakis

**Affiliations:** 1Department of Urology, James Buchanan Brady Urological Institute, Johns Hopkins University School of Medicine, Baltimore, MD 21287, USA; mkanaya1@jhmi.edu (M.K.); clu21@jhmi.edu (C.L.); jluo2@jhmi.edu (J.L.); 2Department of Oncology, Sidney Kimmel Comprehensive Cancer Center, Johns Hopkins University School of Medicine, Baltimore, MD 21287, USA

**Keywords:** androgen receptor splice variants, AR-V7, circulating tumor cells, castration-resistant prostate cancer

## Abstract

**Simple Summary:**

Androgen receptor splice variants (AR-Vs) play an important role in prostate cancer progression, especially as a putative resistance mechanism against AR-targeted therapies. Recent technological advances have enabled detection of AR-Vs in many types of human specimens including circulating tumor cells. Here, we discuss the biology of AR-Vs, the clinical utility of AR-Vs as prognostic and predictive biomarkers, and AR-Vs as potential therapeutic targets with a special focus on AR-V7.

**Abstract:**

Over the past decade, advances in prostate cancer research have led to discovery and development of novel biomarkers and effective treatments. As treatment options diversify, it is critical to further develop and use optimal biomarkers for the purpose of maximizing treatment benefit and minimizing unwanted adverse effects. Because most treatments for prostate cancer target androgen receptor (AR) signaling, aberrations affecting this drug target are likely to emerge following the development of castration-resistant prostate cancer (CRPC), and it is conceivable that such aberrations may play a role in drug resistance. Among the many AR aberrations, we and others have been studying androgen receptor splice variants (AR-Vs), especially AR-V7, and have conducted preclinical and clinical studies to develop and validate the clinical utility of AR-V7 as a prognostic and potential predictive biomarker. In this review, we first describe mechanisms of AR-V generation, regulation and their functions from a molecular perspective. We then discuss AR-Vs from a clinical perspective, focusing on the significance of AR-Vs detected in different types of human specimens and AR-Vs as potential therapeutic targets.

## 1. Introduction

The androgen receptor (AR) is a member of nuclear receptors activated by androgenic ligands such as testosterone and dihydrotestosterone. Because the development and progression of prostate cancer (PCa) depends on AR signaling, inhibition of AR signaling by androgen deprivation therapy (ADT) (e.g., luteinizing hormone-agonist/antagonist with or without casodex) is the mainstay of treatment for advanced castration-sensitive PCa (CSPC). However, CSPC eventually develops resistance to ADT and progresses to castration-resistant PCa (CRPC), often by gaining an ability to activate AR signaling under low-ligand environments. Even after CRPC develops, ADT is continued indefinitely and combined with secondary hormone therapy agents that further block AR signaling either by suppression of androgen synthesis (e.g., abiraterone) or by direct inhibition of AR (e.g., enzalutamide, apalutamide and darolutamide). Because all these drugs target the AR ligand-binding domain (LBD), truncated AR splice variants (AR-Vs) lacking the LBD, i.e., the drug target, have been evaluated as a biologically plausible mechanism of drug resistance to AR-targeting agents. Some AR-Vs have been validated as having constitutively active AR activity in the absence of androgenic ligands [1,2]. Among many AR-Vs, AR-V7 is the most well-studied, and plays an important role in PCa progression and therapy resistance. In this review, we will summarize the biology of AR-Vs and their clinical implications with a special focus on AR-V7.

## 2. Structure of AR-Vs and Mechanisms Underlying Genesis of AR-Vs

The human *AR* gene comprises eight canonical exons, and the full-length AR (AR-FL) protein contains four functional domains: the NH_2_-terminal domain (NTD, encoded by exon 1), the DNA-binding domain (DBD, encoded by exons 2 and 3), the hinge region (encoded by exons 3 and 4), and the ligand-binding domain (LBD, encoded by exon 4–8). The nuclear localization signal (NLS) is located at the junction between the DBD and the hinge region, spanning both exon 3 and 4 [3]. Once bound by ligands in the cytosol, AR releases heat shock proteins, resulting in a conformational change that exposes NLS. Then, AR dimerizes and translocates to nucleus followed by transcriptional activation of AR target genes [3]. Most AR-Vs retain AR NTD and DBD but have LBD truncated. AR-Vs validated in our dataset are summarized in Figure 1 (modified from Figure 1 by Lu et al. [4]). Although many AR-Vs (including AR-V7) lack exon 4, encoding a part of NLS, they may still enter the nucleus and activate transcription to varying degrees independent of the canonical AR NLS and in the absence of AR-FL, possibly due to the NLS-like basic amino acid sequences downstream of AR DBD [1,5]. AR-Vs can be categorized into the following four groups depending on their nuclear localization ability: ligand stimulated in a similar manner to canonical AR-FL (e.g., AR-23), constitutively active (e.g., AR-V3, 4, 7, 12), conditionally active (e.g., AR45, AR-V1, 9), and inactive (e.g., AR-V13, 14, AR8) [4]. AR-Vs reported in other studies but yet to be fully characterized [6,7], and those arising from diverse *AR* gene rearrangements [8], are not included in Figure 1.

Two major mechanisms driving the expression of AR-Vs have been reported in the literature—aberrant RNA splicing intragenic and rearrangements of the *AR* gene [9]. Regulation of AR-V expression by aberrant RNA splicing is considered to be a rapid and reversible adaptive response to ADT rather than a process involving clonal selection, although in many (but not all) cases AR-V expression is accompanied by elevated AR-FL expression which, in turn, is known to be associated with *AR* gene amplification [10,11,12,13]. For example, in LNCaP95 cells derived from LNCaP [14], AR-V7 is acutely induced upon androgen depletion in vitro [12]. Similarly, in a xenograft model of VCaP harboring an increased copy number of wild-type AR [14,15], AR-V7 is induced at as early as 4 days post castration [10]. We will discuss the mechanism of this rapid AR-V induction by aberrant splicing in the next chapter.

Within a subset of PCa, intragenic rearrangements are implicated in the genesis of AR-Vs. The first well-characterized model of *AR* intragenic rearrangements is the castration-resistant CWR22Rv1 cell line. In CWR22Rv1, tandem duplication of a 35-kb segment harboring exon 3 and CEs is linked to AR mRNA including AR-V7 [16]. Similarly, intragenic in-frame deletion or inversion of exon 5, 6 and 7 is responsible for the synthesis of the AR^v567es^ variant in LuCaP 86.2 and LuCaP 136 xenograft, respectively [15,17]. Likewise, an intragenic deletion in intron 1 induces a splicing switch that promotes preferential AR-V7 expression under ADT conditions in the CWR-R1 cell line [15]. Consistent with these in vitro study results [15,16,17], *AR* genomic rearrangements also drive the expression of AR-Vs in human CRPC specimens [8]. *AR* DNA sequencing has revealed that intricate *AR* genomic arrangements are occurring in CRPC specimens, some of which are associated with AR-V expression [8]. Further study is needed to determine a causal relationship between each rearrangement and the expression of AR-Vs.

To summarize, AR-Vs generated either by intragenic rearrangements or aberrant RNA splicing show varying degrees of transcriptional activity and some of them, especially constitutive active AR-Vs such as AR-V7, are likely the result of altered hormonal environment where canonical AR-FL signaling is suppressed as well as the permissive genomic and epigenomic features acquired in the development of CRPC, as we elaborated in a recent review [9].

## 3. AR-V Regulation, Dimerization and Transcriptional Activity

AR-Vs emerge under ADT as an adaptive response to a low-ligand environment [10,11,12]. For induction of AR-Vs by aberrant RNA splicing, several RNA-binding proteins (RBP) that regulate mRNA splicing have been reported to work cooperatively. Liu et al. reported that recruitment of specific splicing factors (U2AF65 and ASF/SF2) to certain splicing enhancers is important for aberrant AR splicing to occur, and enzalutamide has been shown to increase this spliceosome recruitment [11]. Another RBP, Sam 68, has been reported to preferentially increase AR-V7 expression by recruiting other spliceosome components, including U2AF65, to splicing enhancers [18]. PSF also forms a complex with other splicing factors to induce the expression of AR-FL and AR-Vs [19]. Recently, the RNA splicing factor SF3B2 has been identified as a critical determinant of AR-V7 expression. SF3B2 directly binds to CE3 and promote the inclusion of CE3 to AR mRNA [20]. Furthermore, emerging evidence suggests that long non-coding RNAs (lncRNAs) and micro RNAs (miRNAs) are involved in AR-Vs expression by regulating RBPs. For instance, ADT elevates the expression of PCa-specific long non-coding RNA PCGEM1. PCGEM1 interacts with aforementioned U2AF65 splicing factor, which in turn promotes the binding of U2AF65 to AR pre-mRNA, leading to AR-V7 expression [21]. Likewise, interaction of U2AF65 with other CRPC-associated lncRNAs promote AR-V7 expression [22]. While some oncogenic miRNAs have been shown to stabilize AR-FL and AR-Vs [23], the downregulation of tumor-suppressive miRNAs results in the upregulation of AR-FL and AR-V7 via accumulation of hnRNPH1 which is another regulator of alternative splicing [24].

As for negative regulation of AR-FL and AR-Vs, dihydrotestosterone (DHT) is known to suppress the expression of AR-FL and AR-V7 [10,12]. An explanation for this negative regulation mechanism is that ligand-bound AR-FL is known to bind to AR binding site 2 (ARBS2) located in intron 2 and inhibits AR-FL mRNA synthesis via recruitment of lysine-specific demethylase 1 (LSD1) [25]. AR-V7 is considered to be negatively regulated by the same mechanism, but interestingly, DHT treatment results in a greater decrease in AR-V7 expression than AR-FL, indicating a mechanism that preferentially decreases AR-V7 [10].

A few previous studies have investigated how AR-Vs transactivate target genes. Ligand-bound AR-FL dimerizes for DNA binding to chromatin, resulting in the recruitment of cofactors and unlocking the transcriptional machinery [26]. AR-Vs may interact with AR-FL (heterodimers) and each other as well (homodimers). Indeed, AR-V7, AR^v567es^ and AR-FL homodimerize or heterodimerize with each other under ADT conditions [27,28]. Moreover, AR-Vs are reported to enhance AR-FL functions; constitutively active AR-V7 and AR^v567es^ facilitate AR-FL nuclear localization in the absence of androgen and facilitate AR-FL nuclear trafficking even in the presence of enzalutamide [29]. AR^v567es^ also helps stabilize AR-FL by slowing protein degradation [28]. Further, AR-V7 can dimerize with AR-Vs lacking nuclear translocation capabilities by themselves, facilitating their nuclear translocation and the transactivation of target genes [30]. In accordance with these observations, introducing mutations to the dimerization interface of AR-Vs compromises their ability to induce the expression of PSA (prostate-specific antigen, a canonical AR target) as well as UBE2C (ubiquitin-conjugating enzyme E2C, an AR-V-specific target), and attenuates AR-V-mediated castration-resistant growth [27], suggesting that AR-Vs require dimerization for their functions. This is still an evolving area of research and further characterization is necessary to elucidate how AR-Vs mediate their transcriptional functions.

As for the transcriptome regulated by AR-Vs, one key question is whether AR-Vs merely substitute for AR-FL or if they mediate distinctive transcriptional programs. Earlier studies, including ours, suggested that AR-Vs activate a unique repertoire of genes in addition to canonical AR-FL target genes, potentially conferring survival benefit to CRPC cells. In particular, a gene set induced by AR-V7 was enriched for cell-cycle genes [12,31]. Likewise, AR^v567es^ appears to induce a unique proliferative program possibly by activating other growth and survival pathways, such as STAT3 [28]. On the other hand, Li et al. reported that AR-Vs activate a largely overlapping transcriptional program, such as AR-FL, suggesting that previously reported differences between AR-Vs and AR-FL transcriptional programs are only a reflection of AR-Vs’ biphasic signaling output [32]. Together, although the existence of AR-V-specific transcriptional programs are likely to be cell context-specific [33], constitutively active AR-Vs can functionally recapitulate AR-FL in the absence of ligands to some degree and confer survival benefits to PCa.

## 4. Detection of AR-Vs in Human Samples and Clinical Implications

AR-Vs can be utilized as prognostic and potentially predictive markers of resistance to AR-targeted therapies such as abiraterone and enzalutamide. AR-Vs and AR structural aberrations detected in different human specimens are summarized in Table 1, and findings from clinical studies are summarized in Table 2. In our view, prognostic biomarkers provide information about overall outcome irrespective of treatments, whereas predictive biomarkers provide information about treatment response especially using one therapy over another (i.e., treatment-selection biomarkers) [34]. Good examples of prognostic biomarkers are PSA and simple clinical parameters such as disease burden and performance status [35]. As compared to prognostic biomarkers, there are fewer predictive biomarkers for PCa [35]; therefore, the validation of predictive biomarkers for treatment selection is an urgent need. As we discuss in this section, the prognostic value of AR-V7 seems almost unquestionable, but the clinical utility of AR-V7 as a predictive biomarker is still controversial because there have not been enough clinical trials conducted stratifying AR-V7 positive patients into two different treatment arms. Here, we summarize previous studies focusing on the utility of AR-Vs as biomarkers.

Our previous proof-of-concept studies demonstrated that AR-V7 mRNA expression in circulating tumor cells (CTCs) is a prognostic marker in the context of AR-targeting therapies [36,37]. In these studies, CRPC patients with AR-V7 positive CTCs exhibited primary and secondary resistance to next-generation AR-targeted agents, such as abiraterone and enzalutamide, with inferior clinical outcomes in terms of PSA responses, PSA-PFS (PSA progression-free survival), PFS, and OS (overall survival). Of note, in both studies, CTC AR-V7 positivity was associated with other poor prognostic factors, including poor performance status, higher disease burden and higher baseline PSA, which is consistent with Sharp et al.’s findings [38]. However, according to the REMARK (Reporting Recommendations for Tumor Marker Prognostic Studies) guidelines, a new biomarker often has at least a modest association with some other standard prognostic markers, but this association itself does not necessarily undermine a prognostic value of a new biomarker, if the new biomarker maintains some association with clinical outcome after adjusting for standard prognostic variables by multivariate analysis [39]. In fact, CTC AR-V7 status remained an independent predictor of PSA response [37], PFS and OS [40] in multivariate analysis, qualifying CTC AR-V7 as a prognostic marker in the PROPHECY trial (to be detailed below).

A few previous studies evaluated whether CTC AR-V7 can be utilized as a predictive marker by comparing the treatment response of CTC AR-V7 positive patients to two different classes of therapies (taxanes vs. enzalutamide or abiraterone) [47]. For AR-V7 positive patients, taxanes appeared more efficacious with regard to PSA-PFS and PFS, suggesting that CTC AR-V7 can be a predictive marker in this context [47]. Scher et al. conducted a similar study using a different CTC AR-V7 detection method that employed immunofluorescence to localize AR-V7 protein [54,55]. Their results also showed that metastatic CRPC (mCRPC) patients with AR-V7 positive CTCs are likely to benefit more from taxane chemotherapy than AR-targeted therapies [54,55]. These findings provided a clinical rationale for treatment selection based on feasible blood-based CTC AR-V7 testing. The ARMOR3-SV trial (phase III galeterone vs. enzalutamide study) is another example of a randomized controlled trial (RCT) stratifying CTC AR-V7 positive patients into two treatment arms. Unfortunately, the predictive ability of AR-V7 could not be evaluated in this study because of early trial termination due to high censorship for the primary endpoint (radiographic PFS) [48]. Hopefully, future RCTs testing the efficacy of more potent AR inhibitors will validate the predictive value of AR-V7.

The PROPHECY (Prospective Multicenter Validation of Androgen Receptor Splice Variant 7 and Hormone Therapy Resistance in High-Risk Castration-Resistant Prostate Cancer) study combined the above two CTC AR-V7 detection assays: CTC AR-V7 mRNA assay [36,37] and CTC nuclear-specific AR-V7 protein assay [54,55]. In this study, CTC AR-V7 status was tested in men with high-risk mCRPC starting abiraterone or enzalutamide, and the prognostic significance of baseline CTC AR-V7 was validated. As a result, the detection of AR-V7 by both blood-based assays were independently associated with shorter PFS and OS in the context of abiraterone and enzalutamide. Thus, these findings further confirmed that AR-V7 is a prognostic marker associated with worse clinical outcomes in men with mCRPC treated with abiraterone or enzalutamide [40]. The final results of the PROPHECY trial showed that, when these same patients were subsequently treated with taxane chemotherapy, the presence of CTC-derived AR-V7 was not associated with inferior clinical outcomes, suggesting that AR-V7 positive patients may still derive benefit from chemotherapy [49].

AR-Vs are detected not only in CTCs but also in other sample types (Table 1), and can be combined with other biomarkers, such as AR gene aberrations for optimal prognostication. Several studies have shown that comprehensive AR profiling by liquid biopsy potentially improves the prognostic value of these blood-based tests. De Laere et al. successfully detected structural rearrangements of the AR gene from the cell-free DNA (cfDNA) of CRPC patients and combined this with CTC-based AR-V detection and CTC enumeration [41]. Fettke et al. detected AR copy number gains and mutations from cfDNA, together with AR-Vs from cfRNA [42,62]. Targeted cfDNA sequencing of PCa genes enabled the detection of mutations in other driver genes, such as RB1 and MYC, as well as AR copy number gains and mutations [43]. Isolated CTCs can be further analyzed by single-cell sequencing methods for more detailed DNA or RNA profiling [45,46]. AR-V7, detected from plasma-derived exosomal RNA, similarly predicts resistance to abiraterone and enzalutamide [44,56]. Such combinatorial liquid biomarkers are yet to be tested in a prospective fashion.

In tissues, the prognostic value of AR-Vs is slightly more equivocal in comparison with liquid biopsy, in part because many AR-Vs can be detected at varying levels in benign prostate and pre-abiraterone/enzalutamide primary PCa specimens [6,7]. However, given that AR-V expression is a continuous variable rather than a categorical variable, we believe that it is important to evaluate the limit of detection of each measurement platform and set a cut-off value to define positivity. Additionally, expression levels should be evaluated along a continuum of disease progression rather than focusing on a snapshot of qualitative AR-Vs presence at a single timepoint. For example, we have quantified AR-V7 mRNA levels in CSPC and CRPC tissues utilizing RNA in situ hybridization [58]. In line with earlier studies [6,7], some CSPC samples showed a low level of AR-V7 transcript expression. Thus, we set the threshold for positivity and also observed that AR-V7 as well as the AR-V7/AR-FL ratio increase as the disease progresses from CSPC to CRPC. In that study, CRPC biopsies expressing higher AR-V7 were significantly associated with shorter PSA-PFS, confirming AR-V7′s role as a prognostic marker [58]. Consistent with this, at the protein level, Sharp et al. demonstrated CRPC-specific AR-V7 protein expression, and positive protein detection is associated with worse outcomes [63]. Protein detection by IHC may be influenced by detection sensitivity/specificity, especially when different antibodies are used. Chen et al.’s immunohistochemistry-based study demonstrated that high nuclear AR-V7 expression and high nuclear AR-V7/AR-FL ratio were associated with shorter biochemical recurrence-free survival in a relatively large patient cohort [57]. Moreover, the absence of AR-V7 in bone marrow biopsies from mCRPC was associated with better treatment response to enzalutamide in a different study [59]. While most pathological studies mainly focus on AR-V7 because of the limited availability of variant-specific antibodies other than AR-V7, Kohli et al. reported that AR-V9 mRNA is often coexpressed with AR-V7 in mCRPC tissues [50]. In summary, AR-Vs, especially AR-V7, are reliably detected in many types of human biospecimens and can serve as negative prognostic biomarkers. More clinical studies with at least two treatment arms are warranted to determine the value of AR-Vs as predictive biomarkers, but some study results [47,54,55] suggest that CTC-based AR-V7 can serve as a predictive biomarker when considering therapeutic options between AR-targeted therapies vs. taxanes.

## 5. AR-Vs as Therapeutic Targets

Given that AR-Vs are key players in CRPC progression, the development of AR-V-targeting therapies is important. Although taxanes are currently the primary treatment option for AR-V7-positive CRPC patients, new drugs targeting AR-Vs are under development. Novel agents targeting AR-Vs are either directly or indirectly summarized in Table 3.

Niclosamide, an FDA-approved antihelminthic drug, was reported to selectively inhibit AR-V7 protein expression by enhanced protein degradation via a ubiquitin-proteasome pathway in preclinical study [70]. However, phase I trial data showed that the original oral formulation of niclosamide failed to reach minimum effective concentration in serum due to poor oral bioavailability (NCT02532114) [64]. To overcome this absorption issue, reformulated niclosamide was developed and has been shown to exceed the effective concentration with an oral well-tolerated dose in phase Ib trial (NCT02807805) in combination with abiraterone [65]. The phase II portion of this study is currently underway. Although AR-V status was not tested in that study, it would be very informative to evaluate it, considering AR-V-targeting characteristics of this drug and potential efficacy reported so far (50% PSA response in 5/9 patients).

EPI compounds directly target the N-terminal domain (NTD) of AR and suppress AR transcriptional activity by inhibiting protein–protein interactions of AR with other co-activators [71,72]. Because AR-Vs possess NTD, EPI compounds may inhibit AR-Vs as well [73]. EPI-506 entered a phase I/II trial, but the study was terminated because of lack of efficacy and poor oral bioavailability of this agent (NCT02606123). A new formulation of this compound, called EPI-7386 [66], is now in phase I/II clinical testing, both as a monotherapy and in combination with enzalutamide (NCT04421222).

The bromodomain and extraterminal (BET) domain-containing protein 4 (BRD4) interacts with AR-FL and AR-Vs, and their interaction is necessary for transcription activation of *AR* target genes. Thus, disrupting this interaction with BET inhibitors hampers PCa growth [74]. Several BET inhibitors are under development (e.g., GSK525762, ODM-207, ABBV-075, JQ1, OTX015, ZEN-3694, GS-5829, PFI-1) [67,74,75,76,77,78,79], most of which have been shown to suppress AR-V7 in preclinical studies [76,77,78,80]. Among them, ZEN-3694 in combination with enzalutamide demonstrated acceptable tolerability and potential efficacy in phase Ib/IIa clinical trial (NCT02711956) [67].

The histone acetyltransferase paralogues, p300 and CREB-binding protein (CBP), are other important coactivators of AR that can serve as therapeutic targets. p300/CBP mediates histone 3 lysine 27 acetylation (H3K27ac) at enhancer regions and increases chromatin accessibility to facilitate *AR* target gene transcription. Inhibition of p300/CBP by small molecule inhibitors has been shown to suppress PCa growth by several mechanisms, such as abrogation of phosphorylated CREB1 and p300/CBP interaction (compound names: SGC-CBP30 and C646) [81] and inhibition of AR/AR-Vs signaling (compound name: CCS1477) [68]. A Phase I/IIa study of CCS1477 is currently underway to evaluate the safety and efficacy of CCS1477 as a monotherapy and in combination with abiraterone or enzalutamide (NCT03568656) [82]. In addition, NEO2734 is a novel dual inhibitor of BET and CBP/p300. In preclinical studies, NEO2734 has been shown to be efficacious against *SPOP*-mutant PCa which carries abundant BET proteins due to impaired degradation [83] and suppress the expression of non-canonical oncogenic *AR* target genes in enzalutamide-resistant cell lines [84].

Another promising agent is a polo-like kinase 1 (PLK1) inhibitor. PLK1 is a multi-functional serine/threonine kinase that positively regulates critical cell cycle events [85]. In CRPC, PLK1 inhibition has been shown to enhance the efficacy of AR-targeted drugs by several mechanisms, including the suppression of cholesterol biosynthesis, subsequent androgen biosynthesis and downregulation of AR-FL as well as AR-Vs. Specifically, PLK1 acting upstream of the PI3K–AKT–mTOR pathway positively regulates lipid metabolism and subsequent steroid hormone synthesis (including androgens) through SREBP (sterol regulatory element binding protein). PLK1 also induces AR expression under castration-induced oxidative stress via AKT-mediated Twist1 activation. Thus, PLK1 inhibition, in turn, suppresses AR signaling in a pleiotropic manner and it exerts synergistic effects when combined with AR-targeted drugs, especially abiraterone [86]. A phase II clinical trial is currently underway to evaluate the efficacy of onvansertib (PLK1 inhibitor) in combination with abiraterone (NCT03414034), and clinical activity has been observed [87].

In addition to these agents, bipolar androgen therapy (BAT) recently came under the spotlight as a new therapeutic intervention for CRPC [60,61]. As shown in earlier studies, androgens can be a double-edged sword for PCa growth [10,12,88]. In fact, CSPC cells require rigorous regulation of androgen concentration during cell division [89]; supraphysiological exogenous androgen disrupts this tight regulation and causes growth arrest [90]. BAT consisting of testosterone supplementation followed by a washout period targets this vulnerability of PCa by rapidly changing serum androgen concentrations between two polar extremes, which theoretically prevents the re-adaptation of CRPC to androgen concentration in microenvironments. BAT is showing promising results so far in terms of re-sensitization to enzalutamide [60,61]. BAT also led to the conversion from CTC AR-V7 positive to negative, but AR-V7 reverted back to positive after AR-targeted-therapy rechallenge in 83% of patients, indicating that BAT alone may not be enough to eradicate AR-V7 positive clones [60]. Considering the favorable safety profile of BAT, combination therapy of BAT with other therapy, such as poly (ADP-ribose) polymerase 1 inhibition [91], or immune checkpoint blockage [92], may have synergistic effects.

One may also wonder about the efficacy of immune checkpoint inhibitors against PCa. The combination of nivolumab (anti-programmed death 1 antibody) plus ipilimumab (anti-cyto-toxic T-lymphocyte antigen 4 antibody) is known to be especially efficacious against various tumors with a high tumor mutational burden (TMB) [93]. Based on a hypothesis that a subset of aggressive CRPC, similar to those expressing AR-V7, may harbor a higher TMB, AR-V7-positive patients were treated with nivolumab plus ipilimumab (Cohort 1) [51] or in combination with enzalutamide (Cohort 2) in a phase II clinical trial (NCT02601014) [52]. Although there was a trend towards better outcomes in the TMB-high group, nivolumab plus ipilimumab showed limited efficacy in this study irrespective of enzalutamide addition. In the CheckMate 650 trial (nivolumab plus ipilimumab, NCT02985957) enrolling mCRPC patients regardless of AR-V7 status, four patients had complete responses, indicating that immune checkpoint therapy is efficacious for a subset of CRPC patients [69]. Additional studies are needed to determine whether AR-V7 is associated with a higher TMB and whether AR-V7-positive patients may respond differentially to immune checkpoint inhibitors. As another form of immunotherapy, a DNA vaccine encoding the LBD of AR, pTVG-AR (MVI-118), has been shown to be safe and well-tolerated in a recent phase I trial (NCT02411786) [94]. pTVG-AR induces CD8^+^T cell-mediated immune response against CRPC overexpressing *AR* under ADT conditions [95]. However, the efficacy of pTVG-AR against CRPC expressing LBD-lacking AR-Vs is yet to be determined.

Last but not least, ^177^Lu-PSMA-617 radioligand therapy [96] is expected to soon be approved by the FDA as a treatment for mCRPC, considering the positive results of phase III VISION trial (NCT03511664). Kessel et al. evaluated the copy number of CTC AR-V7 mRNA prior to ^177^Lu-PSMA-617 administration [53]. Although the number is small, PSA decline was observed in both AR-V7 negative and positive groups (88% vs. 81%), and there was no statistical difference in PFS and OS between two groups [53]. If this result is validated in larger cohorts, then ^177^Lu-PSMA-617 may be another primary treatment option for AR-V7 positive mCRPC in addition to taxanes.

There are several promising agents directly or indirectly targeting AR-Vs, but we need to keep in mind that AR signaling may not be required in later stages of prostate cancer progression when lineage plasticity emerges following AR-targeting therapies, and AR targeting approaches are generally not considered to be effective in the treatment of neuroendocrine/small cell prostate cancers [97]. From this perspective, some treatments under development may help to slow down the progression to AR-independent prostate cancers. For example, treatments that alleviate selective pressure of ADT, such as BAT or non-AR-related therapy, such as ^177^Lu-PSMA-617 in combination with AR-targeted drugs, may be a good strategy to outsmart PCa’s adaptive ability to its microenvironment.

## 6. Conclusions

Recent technological advances have facilitated AR-V detection in many different types of human specimens and have shed light on the complex biology of constitutively active AR-Vs and their roles in CRPC progression. We believe that AR-V7 is the most important constitutively active AR-V and can be utilized as a prognostic and potential predictive biomarker. Therapeutic targeting of AR-Vs remains our opportunity and challenge for the future.

## Figures and Tables

**Figure 1 cancers-13-02563-f001:**
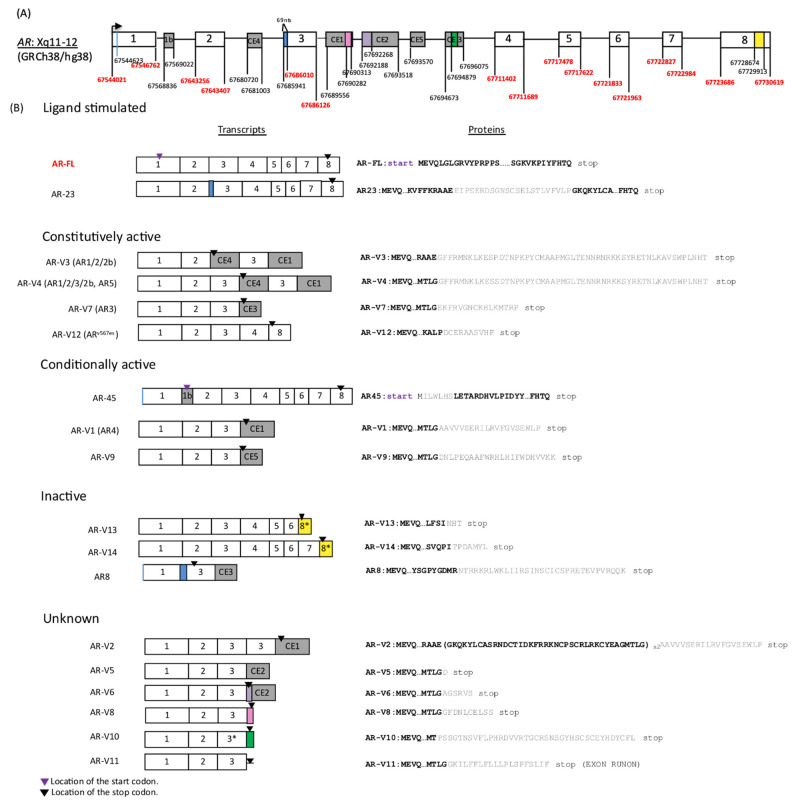
Summary of AR-V structures. (**A**) The structure and locations of *AR* gene exon and cryptic exon (CE) junctions are depicted according to GRCh38/hg38 (not drawn to scale). Non-shaded boxes represent canonical exons and shaded boxes represent exon 1b and CEs. Coordinates corresponding to the canonical exon junctions are labeled with red numbers, whereas black numbers mark the coordinates for exon 1b and CEs as well as locations of other variant-specific sequences in some AR-Vs (AR-23, AR8, AR-V6, 8, 10) shown as color-coded boxes. AR-23 and AR8 contain a 69-nucleotide insertion (blue) immediately upstream of the 5′ junction for Exon 3. AR-V6 has an 80-nucleotide insertion (purple) upstream of 5′ end of CE2. A pink box (3′ end of AR-V8) is a part of CE1 at 3′ end of CE1. A green box (3′ end of AR-V10) is located in the middle of CE3. A yellow box in exon 8 was originally named exon 9. AR-45 and AR8 start from 602 nucleotides downstream of Exon 1 (a blue line). The 3′ end of CE1 shown in this figure is approximately 550 nucleotides downstream of the sequence reported in NCBI database. (**B**) AR-Vs are classified according to their transcriptional activity. Alternative names are shown in brackets. Purple arrowheads indicate locations of a start codon. All AR-Vs, excluding AR-45, have the same start codon as AR-FL; thus, purple arrowheads are omitted from the figure. Black arrowheads indicate the locations of stop codons. AR-V10 has a 7-nucleotide truncation at 3′ end and it is indicated as 3*. A portion of exon 8, indicated as 8* in AR-V13 and 14, was originally called exon 9. Proteins are shown on the right. Amino acids, in bold black font, originate from canonical exons and those in grey are variant-specific peptide sequences.

**Table 1 cancers-13-02563-t001:** AR-Vs and AR structural aberrations detected in human specimens.

Sample Type		AR-Vs and Other AR Structural Aberrations
**Plasma**	**cfDNA**	Intra-AR structural variation [41], Point mutations [41,42,43], Amplification [43], Copy number gain [42,43,44]
	**cfRNA**	AR-V7 [42], AR-V9 [42]
**CTC**	**DNA**	Amplification [45], Copy number loss [45], Copy number gain [45], Deletion [45]
	**RNA**	AR-45 [41], AR-V1 [41,46], AR-V2 [41], AR-V3 [41,46], AR-V4 [46], AR-V5 [41], AR-V7 [36,37,38,40,41,46,47,48,49,50,51,52,53], AR-V9 [41,50], AR-V12 [46], Point mutations [46]
	**Protein**	AR-V7 [40,49,54,55]
**PBMC**	**DNA and RNA**	PBMC is used to detect germline mutations. Because AR mutations in PCa are somatic, PBMC is not used for AR mutation detection. Instead, PBMC is utilized to detect mutations in genes where germline mutations are common such as in BRCA1/2.
**Exosome**	**RNA**	AR-V7 [44,56]
**Tissue**	**DNA**	Amplification [7], Missense mutation [6,7], In-frame indels [7]
	**RNA**	AR-45 [50], AR-23 [50], AR-V1 [6,7], AR-V3 [6,7,50], AR-V5 [6,7], AR-V6 [6,7], AR-V7 [6,7,50,57], AR-V7 (RISH) [58], AR-V8 [6,7], AR-V9 [6,7,50], AR-v5es [6,7], AR-v56es [6,7], AR-v7es [6,7], AR-V13 [6,7], AR-V14 [6,7],
	**Protein**	AR-V7 [19,57,59]

Abbreviations: PBMC—Peripheral blood mononuclear cell.

**Table 2 cancers-13-02563-t002:** AR-Vs as prognostic and predictive markers in CRPC based on clinical studies.

Types of AR-Vs	Significance	Reference
AR-V7	CTC AR-V7 is associated with resistance to ABI and ENZ.	[36,37,40,49]
	Nuclear AR-V7 in CTC is associated with superior survival on taxane chemotherapy over AR-targeted therapy.	[54,55]
	CTC AR-V7 is associated with CTC counts and disease burden. There is sometimes discordance between CTC AR-V7 and tissue AR-V7.	[38]
	CTC AR-V7 is not associated with resistance to taxanes.	[47,49]
	CTC AR-V7 is associated with advanced disease. The ability of AR-V7 to serve as a treatment-selection marker for galeterone could not be evaluated.	[48]
	The presence of any AR-V in CTC is associated with shorter PFS after 2nd hormonal treatment.	[41]
	Exosomal AR-V7 is associated with resistance to ABI and ENZ.	[44,56]
	AR-V7 in biopsies detected by RISH is associated with a shorter PFS.	[58]
	High AR-V7 and AR-V7/ AR-FL ratio in nuclear of PCa tissues are associated with shorter BCR-free survival.	[57]
	AR-V7 in bone marrow is associated with resistance to ENZ.	[59]
	CTC AR-V7 is associated with shorter PFS and OS on BAT and enzalutamide, but not predictive of treatment effects.	[60,61]
	Nivolumab plus ipilimumab showed modest efficacy in CTC AR-V7(+) patients irrespective of enzalutamide addition.	[51,52]
	CTC AR-V7 is not associated with PFS and OS after ^177^Lu-PSMA-617 therapy.	[53]
AR-V1, AR-V2, AR-V3, AR-V5, AR-V7, AR-V9 and AR-45	AR gain and cumulative number of AR aberrations including AR-V7 and AR-V9 are associated with shorter PFS and OS.	[42]
AR-V7 and AR-V9	AR-V9 in tissue is associated with resistance to ABI.	[50]

Abbreviations: ABI—abiraterone acetate, ENZ—enzalutamide, PFS—progression-free survival, OS—overall survival, RISH—RNA in situ hybridization, BCR—biochemical recurrence, BAT—bipolar androgen therapy.

**Table 3 cancers-13-02563-t003:** Therapies targeting AR-Vs.

	Agent	Description	Compound Name, NCT Number and Reference
Directly Targeting AR-Vs	Niclosamide	Inhibit AR-Vs activity by protein degradation. Refomulated niclosamide has improved oral bioavailability.	Niclosamide (NCT02532114) [64] Refomulated niclosamide (NCT02807805) [65]
	EPI compounds	Target N-terminal domain of AR-FL and AR-Vs and suppress transcriptional activity by inhibiting protein–protein interactions of AR-FL and AR-Vs with other co-activators.	EPI-506 (NCT02606123) EPI-7386 (NCT04421222) [66]
Indirectly Targeting AR-Vs	BET inhibitors	Disrupt the interaction between BRD4 and AR-FL and AR-Vs.	ZEN-3694 (NCT02711956) [67]
	CBP/p300 inhibitors	Suppress AR and AR-V7 signaling by inhibiting CBP/p300 (AR coactivators). NEO2734 simultaneously targets BET and CBP/p300.	CCS1477 (NCT03568656) [68]
	PLK1 inhibitors	Inhibit cell cycle progression. Suppress cholesterol biosynthesis and downregulate AR-FL and AR-Vs.	Onvansertib (NCT03414034)
	Bipolar androgen therapy	Supraphysiological exogenous androgen inhibits PCa growth and re-sensitizes CRPC to AR-targeted drugs.	NCT02090114 [60] NCT02286921 [61]
	Immune checkpoint inhibitors	Nivolumab plus Ipilimumab may have modest activity in AR-V7–expressing CRPC patients and/or in those with high TMB.	NCT02601014 [51,52] NCT02985957 [69]
	^177^Lu-PSMA-617	PSMA ligands labeled with β-radiating lutetium-177 target PSMA-expressing PCa cells.	NCT03511664

## Data Availability

Not applicable.

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
