# Peer review of "AR Splicing Variants and Resistance to AR Targeting Agents"

_cancers, 2021, doi:10.3390/cancers13112563_

Round 1
Reviewer 1 Report
The authors gave a clear, comprehensive, and up-to-date review of AR variants in prostate cancer focusing on ARV regulation, functions, and clinical utilities.
Author Response
We sincerely appreciate your kind comments.
Reviewer 2 Report
This review articles by Kanayama and colleagues provide an excellent overview of the current status of the biology of AR-Vs in prostate cancer, the clinical utility of AR-Vs as prognostic and predictive biomarkers, and AR-Vs as potential therapeutic targets with a special focus on AR-V7. The authors described adequate background regarding AR-Vs and discussed important recent progresses and future perspectives. The article is well written and timely.
Author Response

(The authors gave the same response as above.)

Reviewer 3 Report
Kanayama et al. summarized the mechanistic and clinical insights to the emerging role of androgen receptor (AR) variant in advanced prostate cancer. They showed the recent reports regarding the structure of AR variants and the mechanisms generating splicing variants. In addition, they reviewed clinical usage of AR variants as a biomarker for prostate cancer and agents which inhibit AR and variants signals and might have efficacy for advanced prostate cancer.
This review article is very intriguing by summarizing important publications well. However, I think several points should be corrected for the publication.
- The authors described that dysregulation of RNA binding proteins (RBPs) in prostate cancer tissues is one of important mechanisms for the generation of splicing variants. However, there are additional mechanism that modulate the function of these RBPs by non-coding RNA or miRNAs. The representative papers are as followed. Please discuss the involvement of non-coding RNAs. Zhang Z et al. Regulation of androgen receptor splice variant AR3 by PCGEM1. Oncotarget. 2016 Mar 29;7(13):15481-91.
Fletcher CE et al. Androgen receptor-modulatory microRNAs provide insight into therapy resistance and therapeutic targets in advanced prostate cancer. Oncogene. 2019 Jul;38(28):5700-5724.
Takayama KI et al. Identification of long non-coding RNAs in advanced prostate cancer associated with androgen receptor splicing factors. Commun Biol. 2020 Jul 23;3(1):393.
Yang Y et al. Dysregulation of miR-212 Promotes Castration Resistance through hnRNPH1-Mediated Regulation of AR and AR-V7: Implications for Racial Disparity of Prostate Cancer. Clin Cancer Res. 2016 Apr 1;22(7):1744-56.
- Please add Ref.19 to Table 1 (Tissue/Protein/AR-V7). AR-V7 IHC was performed in prostate cancer tissues.
- 286 AR-Vs degradation by niclosamide is an interesting mechanism. Please add more description about this molecular mechanism. Is AR-V7 specifically degraded by ubiquitylation or autophagy? How this small molecule is involved?
- 327 How is PLK1 involved in the regulation of cholesterol biosynthesis or AR expression?
Author Response
Referee #3 (Comments to the Author):
Kanayama et al. summarized the mechanistic and clinical insights to the emerging role of androgen receptor (AR) variant in advanced prostate cancer. They showed the recent reports regarding the structure of AR variants and the mechanisms generating splicing variants. In addition, they reviewed clinical usage of AR variants as a biomarker for prostate cancer and agents which inhibit AR and variants signals and might have efficacy for advanced prostate cancer.
This review article is very intriguing by summarizing important publications well. However, I think several points should be corrected for the publication.
Thank you for evaluating our work. We sincerely appreciate your feedback. We have addressed all concerns as detailed below. Revisions made to the manuscript are indicated in red.
- The authors described that dysregulation of RNA binding proteins (RBPs) in prostate cancer tissues is one of important mechanisms for the generation of splicing variants. However, there are additional mechanism that modulate the function of these RBPs by non-coding RNA or miRNAs. The representative papers are as followed. Please discuss the involvement of non-coding RNAs.
- Zhang Z et al. Regulation of androgen receptor splice variant AR3 by PCGEM1. Oncotarget. 2016 Mar 29;7(13):15481-91.
- Fletcher CE et al. Androgen receptor-modulatory microRNAs provide insight into therapy resistance and therapeutic targets in advanced prostate cancer. Oncogene. 2019 Jul;38(28):5700-5724.
- Takayama KI et al. Identification of long non-coding RNAs in advanced prostate cancer associated with androgen receptor splicing factors. Commun Biol. 2020 Jul 23;3(1):393.
- Yang Y et al. Dysregulation of miR-212 Promotes Castration Resistance through hnRNPH1-Mediated Regulation of AR and AR-V7: Implications for Racial Disparity of Prostate Cancer. Clin Cancer Res. 2016 Apr 1;22(7):1744-56.
We agree with the reviewer that the regulation of RBPs by lncRNAs and miRNA is important and supported by the literature. Therefore, we added above four references and the following context to section 3 of the manuscript:
“Furthermore, accumulating evidence suggests that long non-coding RNAs (lncRNAs) and micro RNAs (miRNAs) are involved in expression of AR-Vs by regulating RBPs. For instance, ADT elevates the expression of PCa-specific long non-coding RNA PCGEM1. PCGEM1 interacts with aforementioned U2AF65 splicing factor, which in turn promotes the binding of U2AF65 to AR pre-mRNA, leading to AR-V7 expression [21]. Likewise, interaction of U2AF65 with other CRPC-associated lncRNAs promote AR-V7 expression [22]. While some oncogenic miRNAs have been shown to stabilize AR-FL and AR-Vs [23], downregulation of tumor-suppressive miRNAs results in upregulation of AR-FL and AR-V7 via accumulation of hnRNPH1 which is another regulator of alternative splicing [24].”.
- Please add Ref.19 to Table 1 (Tissue/Protein/AR-V7). AR-V7 IHC was performed in prostate cancer tissues.
We appreciate your due diligence. Takayama et al.’s reference (reference 19) has been added to Table 1.
- 286 AR-Vs degradation by niclosamide is an interesting mechanism. Please add more description about this molecular mechanism. Is AR-V7 specifically degraded by ubiquitylation or autophagy? How this small molecule is involved?
We agree with the reviewer that it is very intriguing. According to Liu et al. (Ref 64), niclosamide preferentially degrades AR-V7 via ubiquitin-proteasome pathway. However, to the best of our knowledge, why and how niclosamide specifically targets AR-V7 but not AR-FL remains unknown. In the recent study, the same group investigated AR/AR-V7 ubiquitination and degradation, but the role of niclosamide was not studied in this paper (Liu et al., Nature communications, 2018). Thus, we just added “via a ubiquitin-proteasome pathway” to the manuscript as this is currently supported by the literature.
- 327 How is PLK1 involved in the regulation of cholesterol biosynthesis or AR expression?
According to Zhang et al. (Ref 85), PLK1 acting upstream of the PI3K–AKT–mTOR pathway positively regulates lipid metabolism and subsequent steroid hormone synthesis (including androgens) through SREBP. In fact, their findings showed that PLK1 inhibition resulted in downregulation of SREBP and decreased testosterone production. As for AR regulation by PLK1, presumably there are several mechanisms involved. According to the paper, one possible mechanism is the activation of Twist1 by PLK1. Specifically, PLK1 activates AKT, then AKT phosphorylates and activates Twist1. Twist1 is known to increase AR expression under castration-induced oxidative stress. Authors found that PLK1 inhibition sequestered Twist1 in cytosol, resulting in decreased AR mRNA expression. However, on the protein level, PLK1 inhibition alone was not sufficient to significantly decrease AR-FL or AR-V7, but once combined with AR-targeted drugs, especially abiraterone, a PLK1 inhibitor efficiently abrogated AR-FL and AR-V7 expression. Thus, in the revised manuscript we elaborated on the PLK1 inhibition mechanism as follows:
“Specifically, PLK1 acting upstream of the PI3K–AKT–mTOR pathway positively regulates lipid metabolism and subsequent steroid hormone synthesis (including androgens) through SREBP (sterol regulatory element binding protein). PLK1 also induces AR expression under castration-induced oxidative stress via AKT-mediated Twist1 activation. Thus, PLK1 inhibition suppresses AR signaling in a pleiotropic manner and it exerts synergistic effects when combined with AR-targeted drugs, especially abiraterone”.
